# Adding the Mureş River Basin (Transylvania, Romania) to the List of Hotspots with High Contamination with Pharmaceuticals

**Alexandru Burcea [1], Ioana Boeraş [1], Claudia-Maria Mihuţ [1], Doru Bănăduc [1,\*], Claudiu Matei [2] and Angela Curtean-Bănăduc [1]**

[1] Applied Ecology Research Center, Faculty of Sciences, Lucian Blaga University of Sibiu, RO-550012 Sibiu, Romania; alexandru.burcea@ulbsibiu.ro (A.B.); ioana.boeras@ulbsibiu.ro (I.B.); claudia.mihut@ulbsibiu.ro (C.-M.M.); angela.banaduc@ulbsibiu.ro (A.C.-B.)

[2] Faculty of Medicine, Lucian Blaga University of Sibiu, RO-550012 Sibiu, Romania; claudiu.matei@ulbsibiu.ro

[\*] Correspondence: ad.banaduc@yahoo.com or doru.banaduc@ulbsibiu.ro; Tel.: +40-722-604-338

**Abstract:** *Background*: The Mureș River Basin is a long-term heavily polluted watershed, in a situation of climate changes with decreasing water flow and related decreasing dilution capacity. Here, a mixture of emerging pollutants such as pharmaceuticals were targeted to reveal potential risks regarding the natural lotic ecosystems. Due to the continuous discharge into the environment, pharmaceuticals are gaining persistent organic pollutant characteristics and are considered emerging pollutants. Based on the hazard quotient, this research highlights the dangerous concentrations of carbamazepine, ibuprofen, furosemide, and enalapril in river water. *Results*: High levels of four pharmaceutical compounds (carbamazepine, ibuprofen, furosemide, and enalapril) and some of their derived metabolites (enalaprilat, carboxyibuprofen, 1-hydroxyibuprofen, and 2-hydroxyibuprofen) were reported in our study in the Mureș River Basin. Overall, pharmaceutical concentrations were found to be highest in the wastewater treatment plant (WWTP) effluent, median downstream of the WWTP, and lowest upstream of the WWTP, as was expected. For all pharmaceutical compounds tested, we recorded concentrations above the limit of quantification (LOQ) in at least one of the sites tested. Carbamazepine exhibited the highest mean values upstream, downstream, and at the WWTP. As expected, the highest concentrations for all the studied pharmaceutical compounds were detected in the WWTP effluent. All Hazard Quotient (HQ) values were below one (on a logarithmic scale in base 10), with the highest values in the WWTP and the lowest in the river upstream of the WWTP. The HQ intervals were in the same range for furosemide, carbamazepine, and ibuprofen at each of the three different sites: upstream WWTP effluent, and downstream. The interval for enalapril stands out as having the lowest HQ at all three sites. *Conclusions*: Based on these results, the large and complex hydrographical system Mureș River Basin was transformed from a grey area, with little information about pharmaceutical contamination, to a hotspot in terms of contamination with emerging pollutants. Pharmaceutical compound concentrations were found to be the highest in WWTP effluents. The WWTP effluent concentrations were among the highest in Europe, indicating that treatment plants are the primary source of water pollution with pharmaceuticals compounds. The detected levels were higher than the safety limit for carbamazepine and ibuprofen. The determined HQ values imply that the measured levels do pose a threat to the environment for the studied pharmaceuticals. Based on the obtained results, human communities can assess, monitor, predict, and adapt in time to these already-present regional challenges and risks for sustainable use of natural resources, including water and associated products and services.

**Keywords:** wastewater treatment plants; emerging pollutants water contamination; hazard quotient; carbamazepine; ibuprofen; furosemide; enalapril; liquid chromatography; triple quadrupole mass spectrometry

---

## 1. Background

Pharmaceuticals are chemical compounds prepared or dispensed in pharmacies and hospitals and used in the medical treatment of humans and animals. They come in the form of prescription, over the counter, veterinary, or therapeutic pharmaceuticals. Due to industrialized production, there are many pharmaceuticals that became relatively largely accessible worldwide on the free market, including in terms of prices [1,2]. In the last decade, the unintentional presence of pharmaceuticals in the aquatic ecosystems (water, sediment, and biota) has become increasingly apparent in concentrations that can have a negative impact on the aquatic organisms and ecological processes. Due to their presence in the environment, pharmaceuticals are starting to be considered emerging pollutants: compounds not yet included in water-quality regulations, with unknown or poorly understood effects, and pose a potential threat to the ecosystems and human safety and health [3].

In the Lower Danube Basin, footprints of human presence go back in history to 180,000 BC, with noticeable increasing damaging effects on the environment throughout time [4–6]. One of the large-scale second-order tributaries of the Danube is the Mureș River; its upper and middle sectors are located in the amphitheater-like Transylvanian depression, ringed by the South-Eastern Carpathians, and inhabited by over seven million people [7], making it a zone containing important human activities causing adverse effects [8].

The Mureș River is the largest tributary of the Tisza River (Danube Basin), with a length of 761 km and a watershed with an area of 28,319 km$^2$, located in the central and western part of Romania (longitude: 20°11′ west and 25°44′ east and latitude: 45°14′ south and 47°08′ north). This basin relief varies significantly; mountains cover 25% of the surface, while 55% of the surface consists of hills and plateaus, 15% valleys and meadows, and 5% plains [9,10].

The Mureş Basin was chosen for this study for several reasons: the relatively large surface/importance in the Danube Basin [6], the relatively significant human population living in the basin area including in large cities [9,11], the important historical and present human impact including pollution problems in the basin [12,13], the presence of numerous WWTPs (wastewater treatment plants) along the river using similar technology (all 15 WWTPs included in this study have mechanical, biological, and chemical processes used for the treatment of wastewater), the diversity of habitats in the watershed, the total lack of pharmaceutical-aquatic ecosystems related data in this basin, etc.

Pharmaceutical compounds have been investigated and found in the sewerage system's contaminated waters, permanently affected by effluents from hospitals, residential, office, and production areas of Mureş River and its main tributaries. These compounds come from different pharmaceutical classes, such as non-steroidal anti-inflammatory (ibuprofen), psychotropic (carbamazepine), cardiovascular (enalapril), and diuretic pharmaceuticals (furosemide). Besides their presence in water environments, these pharmaceuticals have been characterized according to their water solubility, predicted no-effect concentration (PNEC), and pKa or log $K_{ow}$ (Table 1). Alongside the parent pharmaceuticals, it is interesting to look at their metabolites. For example, enalapril and the ibuprofen metabolites (carboxyibuprofen, 1-hydroxyibuprofen, and 2-hydroxyibuprofen) can be found in the environment. Of the above-listed compounds, carbamazepine is identified as a future emerging pollutant priority candidate, while ibuprofen is a proposed addition to this list [14]. One reason carbamazepine has been thoroughly investigated is its ubiquitous presence (94%) in analyzed rivers. This presence is not so much due to its high use but more likely due to its slow degradation rate and ability to be extracted efficiently from contaminated samples or efficient extraction methods [15,16].

**Table 1.** Properties of the studied pharmaceuticals.

| Substance | CAS Number | Molecular Weight | Water Solubility | pKa | Log $K_{ow}$ | PNEC |
|---|---|---|---|---|---|---|
| Enalaprilat | 76420-72-9 | 348.399 g/mol | 0.876 mg/mL | 3.13 | −0.94 [17] | NA |
| Enalapril | 75847-73-3 | 376.453 g/mol | 16.4 g/L [18] | 3.67 | 4.22 | 184 μg/L [19] |
| Furosemide | 54-31-9 | 330.739 g/mol | 73.1 mg/L [20] | 4.25 | 2.03 | 6.2 μg/L [19] |
| Carbamazepine | 298-46-4 | 236.274 g/mol | 0.152 mg/mL | 15.96 | 2.45 | 7.7 μg/L [19] |
| Ibuprofen | 15687-27-1 | 206.285 g/mol | 21 mg/L [20] | 5.3 [21] | 3.97 | 2.3 μg/L [19] |
| Carboxyibuprofen | 15935-54-3 | 236.267 g/mol | 0.3 g/L | 3.97 | 2.78 [22] | NA |
| 1-hydroxyibuprofen | 53949-53-4 | 222.284 g/mol | 0.51 g/L | 4.55 | 2.69 [22] | NA |
| 2-hydroxyibuprofen | 51146-55-5 | 222.284 g/mol | 0.3 g/L | 4.63 | 2.37 [22] | NA |

CAS—chemical abstracts service. pKa—acid dissociation constant. Log $K_{ow}$—octanol/water partition coefficient. PNEC—predicted no-effect concentration. NA—not available.

## 1.1. Pollution Sources and Environmental Hazards of Studied Pharmaceutical Compounds

It has been reported that some pharmaceuticals that are present in surface waters, groundwater, and the discharge from WWTPs pose a severe environmental problem, since these compounds could affect non-target and susceptible species because they are biologically active [23,24]. Furthermore, these compounds have potentially toxic effects (or could determine behavioral alteration) in the aquatic trophic nets, affecting the food chain organisms such as phytoplankton [25], amphipods [26] and crustaceans [27], fish [28,29], and finally, mammals [30]. When looking specifically at the pharmaceuticals of interest, it is observed that carbamazepine induces a stress response in rainbow trout individuals (*Oncorhynchus mykiss*) [28]. At the same time, ibuprofen negatively affects the health of African catfish (*Clarias gariepinus*) individuals [29].

Due to their polar nature (Table 1), these compounds stay in the solution and do not adhere to soil and particles; therefore, they are mobile in the environment. Another downside of pharmaceuticals is the continuous release into the aquatic environment, which gives them the characteristic of persistent organic pollutants (POPs) regarding their high detection rate. These characteristics make the studied pharmaceutical compounds likely to reach drinking water sources, posing a serious problem for human safety and health in places dependent on recycled water. The problem has been reported in France [23], the United States [31], and Australia [32]. In Romania, the effluent from WWTPs is not reused as drinking water and is discharged back into rivers [33]. This effluent mixes with the hyporheic water and can potentially influence other downstream water sources. When considering the dilution of the pharmaceuticals, the human risk is lowered, but there are still problems when mixtures are involved [34], and new compounds are added to the mixture every day. On the other hand, the threat to the environment is problematic, and high interest is accorded to pharmaceutical's presence and their effects on flora and fauna [35]. From this point of view, the studied basin is a grey area, with few reports about pharmaceutical concentrations [36–38], a relatively common situation, especially in the southeastern part of Europe, but not only.

The two primary biological sources of pharmaceuticals in the studied environment are derived from veterinary and medical uses, through animal and human excretion of active metabolites consisting of a mixture of metabolized and conjugated compounds and unmetabolized compounds [3]. Humans excrete mainly 55 to 80% unmetabolized compounds (with few exceptions) through urine and partially through feces [39–41]. The following can be different sources of aquatic environment contamination: WWTP discharges, hospital effluents, direct disposal of unused or expired pharmaceuticals, manufacturing, landfill leachates, livestock activities, aquaculture, and soil fertilization with sewage sludge and livestock waste [42–44]. Among the mentioned sources, WWTPs are considered the most problematic source of contamination [45]. This is because the fact that WWTPs do not effectively eliminate all the pharmaceutical compounds from the WWTP influents during the procedures of removing pollutants [46–58]. Examples of inadequate removal of pharmaceuticals at the WWTP are carbamazepine, with a low (10–20%) to no removal efficiency [53,54], and furosemide, with a removal efficiency under 42% [55]. Pharmaceuticals could have long-term effects on biota and could exhibit bioaccumulation, and the presence of different pharmaceutical mixtures, which could have additive and

synergistic effects [3]. In terms of avoiding pharmaceutical contamination of water, a series of methods have been proposed and implemented, such as physical adsorption processes, biological degradation processes, chemical processes, advanced oxidation processes, and various combined methods [56]. Alongside these methods, a series of procedures limiting the disposal of surplus pharmaceuticals should also be considered.

Very few studies investigated the occurrence of pharmaceuticals in river waters in Romania [57–61]; even fewer studies mention parts of the investigated area and pharmaceuticals [36–38].

*1.2. Aim*

This study focuses on the occurrence, distribution, and fate of several pharmaceuticals that are on the emerging pollutants list or that are commonly used in the Mureș River Basin, where no large-scale data about them are widely available. This study aims to assess the potential risk of the investigated pharmaceuticals in the environment. One way of tackling this problem is using the hazard quotient (HQ: ≥1 indicates a potential for negative impact on the lotic ecosystem, <1 low ecological risk), i.e., the ratio between the measured environmental concentration (MEC) and predicted no-effect concentration (PNEC). HQ is the measure of the potential exposure to a substance for which no adverse effect is expected.

## 2. Material and Methods

*2.1. Sampling*

Water samples were collected from 15 sites in the Mureș River Basin in 2018. The sampling sites are located from the river source to the Mureș River exit from the Romanian territory (Figure 1). The Mureș River is located in the central and western part of Romania (longitude: 20°11′ west and 25°44′ east and latitude: 45°14′ south and 47°08′ north). Sites were selected around WWTPs: upstream, downstream, and at the WWTPs effluents. The distance from the upstream site and WWTP effluent or the downstream site and WWTP effluent were 100 m. For each site, three replicates were collected. Water was collected in plastic (polyethylene terephthalate) bottles, capped, and maintained at 4 °C for a maximum of two weeks. Pharmaceuticals of interest were extracted from the water samples.

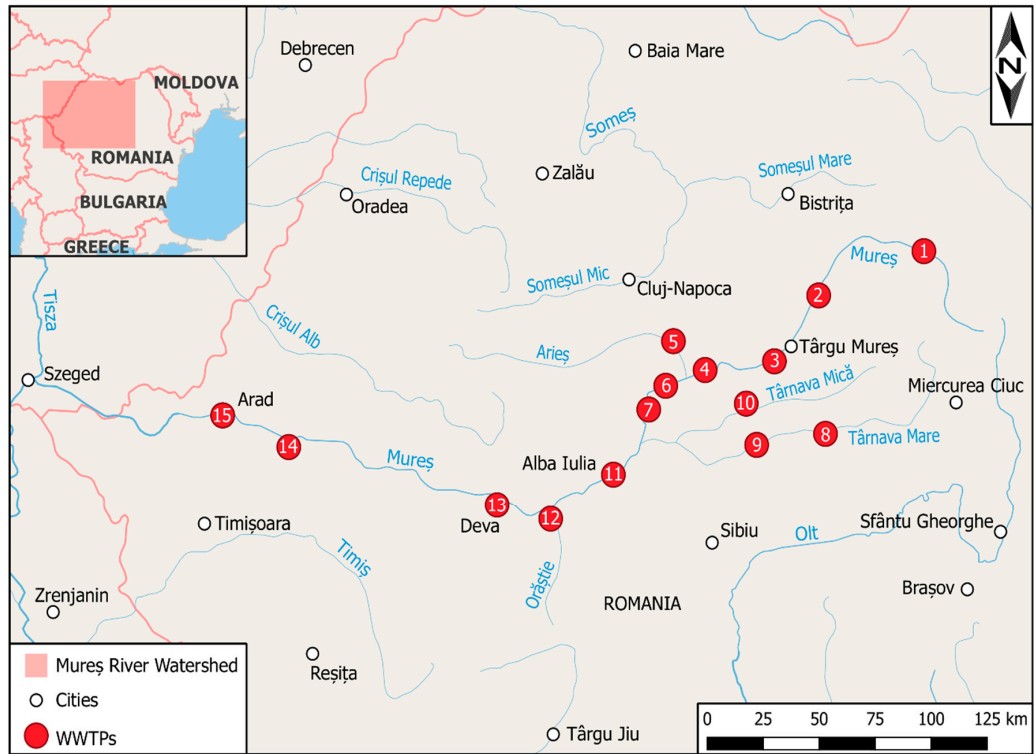

**Figure 1.** Map showing sampling locations at the wastewater treatment plant (WWTP) effluent: 1—Toplița (46°56′5.348″ north, 25°19′9.469″ east), 2—Reghin (46°45′19.458″ north, 24°42′35.128″ east), 3—Cristești (46°29′57.012″ north, 24°27′46.418″ east), 4—Luduș (46°27′22.9″ north, 24°3′42.274″ east), 5—Câmpia Turzii (46°33′40.518″ north, 23°52′18.372″ east), 6—Ocna Mureș (46°23′25.08″ north, 23°50′11.396″ east), 7—Aiud (46°17′50.1″ north, 23°44′36.799″ east), 8—Sighișoara (46°13′44.818″ north, 24°46′16.341″ east), 9—Mediaș (46°10′43.403″ north, 24°22′29.74″ east), 10—Târnăveni (46°20′3.826″ north, 24°18′25.189″ east), 11—Alba Iulia (46°2′.657″ north, 23°33′15.112″ east), 12—Orăștie (45°51′41.152″ North, 23°12′17.798″ East), 13—Deva (45°54′15.869″ north, 22°53′36.243″ east), 14—Lipova (46°4′35.256″ north, 21°40′38.161″ east), 15—Arad (46°10′40.152″ north, 21°16′37.829″ east).

## 2.2. Reagents

All the reagents were purchased from BioAqua (Târgu Mureș, Romania) and were of HPLC analytical grade (water, acetonitrile, and methanol), except for the formic and acetic acids of 98% purity. Carbamazepine (99.5%) and 10,11-dihydrocarbamazepine (99,5%) were obtained from Dr. Ehrenstorfer reference standards producer (Germany), enalapril (97%) and furosemide (97%) were purchased from Alfa Aesar (Lancashire, UK), enalaprilat (98%) was obtained from Cayman Chemical Company (Ann Arbor, MI, USA), and ibuprofen (98%), carboxyibuprofen (98%), 1-hydroxyibuprofen (98%), 2-hydroxyibuprofen (98%), and ibuprofen-d3 (98%) were purchased from Merck (Darmstadt, Germany). The Oasis HLB 500 mg cartridges were purchased from Waters (Wilmslow, UK).

## 2.3. Sample Extraction

Compounds of interest were extracted using the solid phase extraction (SPE) method on 500 mg Oasis HLB cartridges [44] installed on a vacuum manifold (Phenomenex) and conditioned with 5 mL of HPLC grade methanol at 5 mL/min. The cartridges were then equilibrated with 5 mL of 0.1% (acetic acid) acidulated HPLC grade water at 5 mL/min. Afterward, 1 L of acidulated (0.1% acetic acid) river water sample was passed through the cartridge at 10 mL/min. After the sample passage, the sample bottles were rinsed with 10 mL HPLC grade water flowed through the cartridges. A full vacuum was used for 30 min to dry the cartridges. The elution was done first with 5 mL of HPLC grade methanol and then with 5 mL of a 1:1:1 solution of methanol:acetonitrile:isopropanol. The eluate was

concentrated to 0.5 mL under a gentle stream of air at 40 °C and stored in 1.5 mL clean vials (Agilent) until analysis.

## 2.4. Liquid Chromatography

Compounds of interest were separated with a 1200 HPLC (Agilent) on a Zorbax SB-C18 (2.1 × 100 mm, 3.5 μm, Agilent) HPLC column [48]. Pharmaceuticals were divided into two groups, and each group was analyzed by a different method. The first group contained: carbamazepine, 10,11-dihydrocarbamazepine, furosemide, enalapril, and enalaprilat. For this group of pharmaceuticals, the mobile phases were HPLC grade water (solvent A) and HPLC grade acetonitrile (solvent B), both with 0.1% formic acid. The flow rate was set to 0.8 mL/min, and the mobile phase gradient was set to rise from 5% solvent B to 15% B in 3 min, held at 15% B for 1 min, raised to 35% B in 3 min, and then raised to 70% B in 3 min with 5 μL injection volume. The second group of pharmaceuticals contained: ibuprofen, carboxyibuprofen, 1-hydroxyibuprofen, 2-hydroxyibuprofen, and ibuprofen-d3. For this group, the mobile phases were HPLC grade water (solvent A) with 0.1% acetic acid and HPLC grade methanol (solvent B). The flow rate was set at 0.8 mL/min with the mobile phase gradient set to rise from 5% solvent B to 100% solvent B in 8 min with a 2 min hold at 100%, and the injection volume was set to 10 μL. The retention times (RT) are presented in Table 2.

## 2.5. Triple Quadrupole Mass Spectrometry

For triple quadrupole mass spectrometry (QqQ MS) [51], we used a G6410B system (Agilent) with a multimode ESI-APCI source installed. The capillary was set to 4 kV for positive mode and 2.5 kV for negative mode. We have used nitrogen for drying (325 °C at 5 L/min) and nebulizing (40 psi). The MS heaters were set at 100 °C, both while nitrogen was used for collision-induced dissociation (CID). All pharmaceuticals were first analyzed in SCAN mode to determine the retention time and then in product ion mode to determine the best fragmentor voltage and collision energy. The MS/MS was operated in multiple reaction monitoring (MRM) mode to detect and quantify the compounds of interest (Table 2). For each pharmaceutical, the most abundant ion transition was selected for quantification (Q), while a second ion transition was chosen for qualification (q), where possible.

**Table 2.** Details regarding the column separation and detection of investigated pharmaceuticals.

| Substance | RT | Ion Mode | Ion Transition | Fragmentor (V) | Collision Energy (V) |
|---|---|---|---|---|---|
| Enalaprilat | 3.808 | Positive | Q 349.2→206.2 | 135 | 15 |
|  |  | Positive | q 349→303.2 | 135 | 12 |
| Enalapril | 7.201 | Positive | Q 377.3→234.2 | 70 | 17 |
|  |  | Positive | q 377.3→303.3 | 70 | 12 |
| Furosemide | 7.593 | Negative | Q 329→285.1 | 120 | 10 |
|  |  | Negative | q 329→204.7 | 110 | 20 |
| Carbamazepine | 7.917 | Positive | Q 237→194 | 110 | 15 |
|  |  | Positive | q 237→191.9 | 110 | 35 |
| * 10,11-dihydrocarbamazepine | 8.052 | Positive | Q 239.1→196 | 110 | 25 |
|  |  | Positive | q 239.1→180.1 | 110 | 25 |
| Ibuprofen | 7.240 | Negative | Q 205→161 | 75 | 5 |
| Carboxyibuprofen | 5.516 | Negative | Q 235→191 | 75 | 5 |
| 1-hydroxyibuprofen | 5.811 | Negative | Q 221→159 | 75 | 5 |
| 2-hydroxyibuprofen | 5.526 | Negative | Q 221→177 | 75 | 0 |
| * Ibuprofen-d3 | 7.239 | Negative | Q 208→164 | 75 | 0 |

Q—quantifier transition. q—qualifier transition. RT—retention time. * internal standards.

LOD (limit of detection) and LOQ (limit of quantification) were determined from the standard deviation of 10 replicate injections, where LOD was three times the standard deviation. At the same time, LOQ was calculated as 10 times the standard deviation (Table 3) [62,63]. The standard curve linearity ($R^2$) was higher than 0.998 for all the analyzed pharmaceuticals.

**Table 3.** Instrument LOD and LOQ values for the studied pharmaceuticals.

| Substance | Quantification Transition | LOD ppb | LOQ ppb |
|---|---|---|---|
| Enalaprilat | 349.2→206.2 | 0.592 | 1.972 |
| Enalapril | 377.3→234.2 | 0.625 | 2.084 |
| Furosemide | 329→285.1 | 0.868 | 2.894 |
| Carbamazepine | 237→194 | 0.355 | 1.183 |
| Ibuprofen | 205→161 | 0.806 | 2.687 |
| Carboxyibuprofen | 235→191 | 0.795 | 2.648 |
| 1-hydroxyibuprofen | 221→159 | 0.736 | 2.453 |
| 2-hydroxyibuprofen | 221→177 | 0.686 | 2.287 |

LOD—limit of detection. LOQ—limit of quantification.

### 2.6. Data Analysis

The concentrations were calculated relative to the internal standards. For carbamazepine, enalapril, enalaprilat, and furosemide, we used 10.11-dihydrocarbamazepine as the internal standard. While ibuprofen-d3 was used as the internal standard for ibuprofen, carboxyibuprofen, 1-hydroxyibuprofen, and 2-hydroxyibuprofen. The arithmetic mean of the three replicates was used for statistical analyses. We have investigated the data distribution by employing the Shapiro-Wilk normality test implemented in the base R 3.5.2 package. A Spearman's rank-order correlation was used to assess the correlation between treated water from the WWTP on river water quality, while the corrplot package [64] was used to generate the correlograms using R 3.5.2 package. GraphPad Prism version 6.0 was used for designing concentration and HQ graphs. The HQ was quantified as the ratio between the MEC of the pharmaceutical and PNEC. The PNEC values used are listed in Table 1. The map (Figure 1) was generated using QGIS 3.6 software [65], the Natural Earth Data maps, and the WGS 84 sampling site coordinates.

### 2.7. Method Validation

To test our ability to extract the selected pharmaceuticals from river water and determine the method's precision and accuracy, eight recovery experiments were undertaken. For this, we sampled 10 L of water from a single location and distributed the water into two blinds and eight recoveries of 1 L each. River water samples (recoveries) were spiked with concentrations close to those expected in the river and extracted using the specified SPE method. River water that was not spiked (blinds) with pharmaceuticals served as a control for the pharmaceutical amount in the environment and was used to subtract the native contamination from the recoveries. The mean recoveries are as follows: 99% for enalaprilat, 101% for enalapril, 99% for furosemide, 109% for carbamazepine, 98% for carboxyibuprofen, 97% for 2-hydroxyibuprofen, 100% for 1-hydroxyibuprofen, and 97% for ibuprofen. The relative standard deviation (RSD) was also calculated, and the values are as follows: 11% for enalaprilat, 8% for enalapril, 14% for furosemide, 4% for carbamazepine, 9% for carboxyibuprofen, 1% for 2-hydroxyibuprofen, 2% for 1-hydroxyibuprofen, and 3% for ibuprofen. Negative controls (solvent blanks) were analyzed at each extraction and quantification to rule out possible contamination of solvents with the tested pharmaceuticals and the presence of background noise acquired during detection and quantification. No compounds of interest were detected in the blinds.

## 3. Results

### 3.1. Concentrations

Concentrations of four pharmaceuticals, enalapril, furosemide, carbamazepine, and ibuprofen, as well as those of some of their metabolites, enalaprilat for enalapril and carboxyibuprofen, 1-hydroxyibuprofen and 2-hydroxyibuprofen for ibuprofen, were determined in river water upstream, downstream, and in the WWTP effluent for 15 WWTPs along the Mureș River Basin. All the reported concentrations were higher than the LOQ. Overall, pharmaceutical concentrations are highest in

the WWTP effluent, median downstream of the WWTP, and lowest upstream of the WWTP, as was expected (Table 4, Table 5).

We next analyzed the minimum quantifiable, maximum, median, and average concentrations measured at the 15 locations tested for each pharmaceutical upstream, downstream, and at the WWTP (Table 4). For all pharmaceuticals tested, we measured concentrations above the LOQ in at least one of the sites tested (Table 5). Carbamazepine exhibited the highest average measured upstream, downstream, and at the WWTP. As expected, the highest concentrations for all the pharmaceuticals were detected in the WWTP effluent (Table 4).

### 3.2. Correlations

We found a strong, positive correlation between upstream WWTP concentrations and downstream concentrations for enalaprilat and furosemide, which was statistically significant ($r_s$ (2) = 0.94, $p < 0.01$ for enalaprilat and $r_s$ (3) = 1, $p < 0.001$ furosemide). In case of furosemide, carboxyibuprofen, and ibuprofen, a moderate, positive correlation between WWTP effluent concentrations and downstream concentrations was determined ($r_s$ (108) = 0.62, $p < 0.05$ for furosemide, $r_s$ (80) = 0.72, $p < 0.01$ for carboxyibuprofen, and $r_s$ (8) = 0.85, $p < 0.05$ for ibuprofen) (Figure 2). For the rest of the investigated relevant pairing of pharmaceuticals, no correlation between upstream, downstream, or WWTP effluent concentrations was determined.

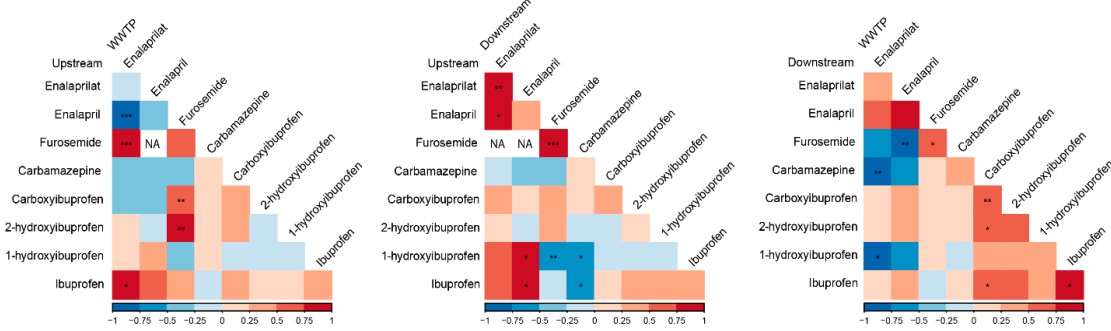

**Figure 2.** Spearman's rank-order correlograms highlight the pharmaceutical correlations between locations. Positive (1) and negative (−1) correlations are represented as a color gradient, while the p values are represented with asterisks (*** $p < 0.001$, ** $p < 0.01$, * $p < 0.05$). NA depicts situations where there were not enough values to compute the correlation.

We also measured correlations between pharmaceuticals and their metabolites at the different sampling sites. We found a strong, positive correlation between enalapril and enalaprilat at each of the three sites, upstream, downstream, and WWTP effluent ($r_s$ (4) = 1, $p < 0.001$ upstream, $r_s$ (12) = 0.78, $p < 0.05$ downstream, $r_s$ (10) = 0.82, $p < 0.05$ in the WWTP effluent). For ibuprofen and its metabolites, a positive correlation has been observed in the following situations: ibuprofen and carboxyibuprofen in the upstream sites ($r_s$ (44) = 0.8, $p < 0.01$), ibuprofen and carboxyibuprofen in the downstream sites ($r_s$ (68) = 0.69, $p < 0.05$), ibuprofen and 1-hydroxyibuprofen in the upstream sites ($r_s$ (116) = 0.59, $p < 0.05$), ibuprofen and 2-hydroxyibuprofen in the upstream sites ($r_s$ (106) = 0.62, $p < 0.05$), ibuprofen and 2-hydroxyibuprofen in the downstream sites ($r_s$ (22) = 0.9, $p < 0.001$), and finally, ibuprofen and 2-hydroxyibuprofen in the WWTP effluent sites ($r_s$ (10) = 0.82, $p < 0.05$) (Figure 3).

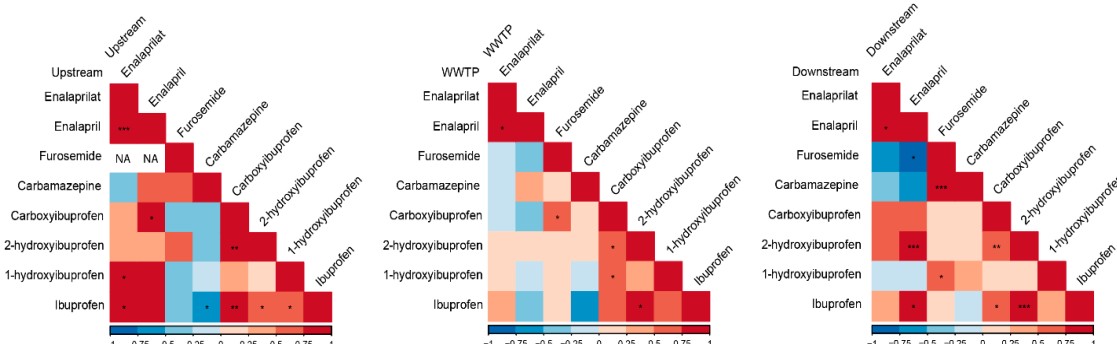

**Figure 3.** Spearman's rank-order correlograms highlight the pharmaceutical correlations within locations. Positive (1) and negative (−1) correlations are represented with color gradients, while the p values are represented with asterisks (*** $p < 0.001$, ** $p < 0.01$, * $p < 0.05$). NA represents situations where there were not enough values to compute the correlation.

## 3.3. Hazard Quotient (HQ)

To address the impact of the investigated pharmaceutical compounds on the aquatic ecosystem, we quantified the HQ for the four pharmaceuticals tested: enalapril, furosemide, carbamazepine, and ibuprofen (Figure 4). All HQ values are below one (on a logarithmic scale in base 10), with the highest values in the WWTP effluent and the lowest in the river upstream of the WWTP. The HQ intervals were in the same range for furosemide, carbamazepine, and ibuprofen at each of the three different sites: upstream, WWTP effluent, and downstream. The interval for enalapril stands out as having the lowest HQ at all three sites (Figure 4).

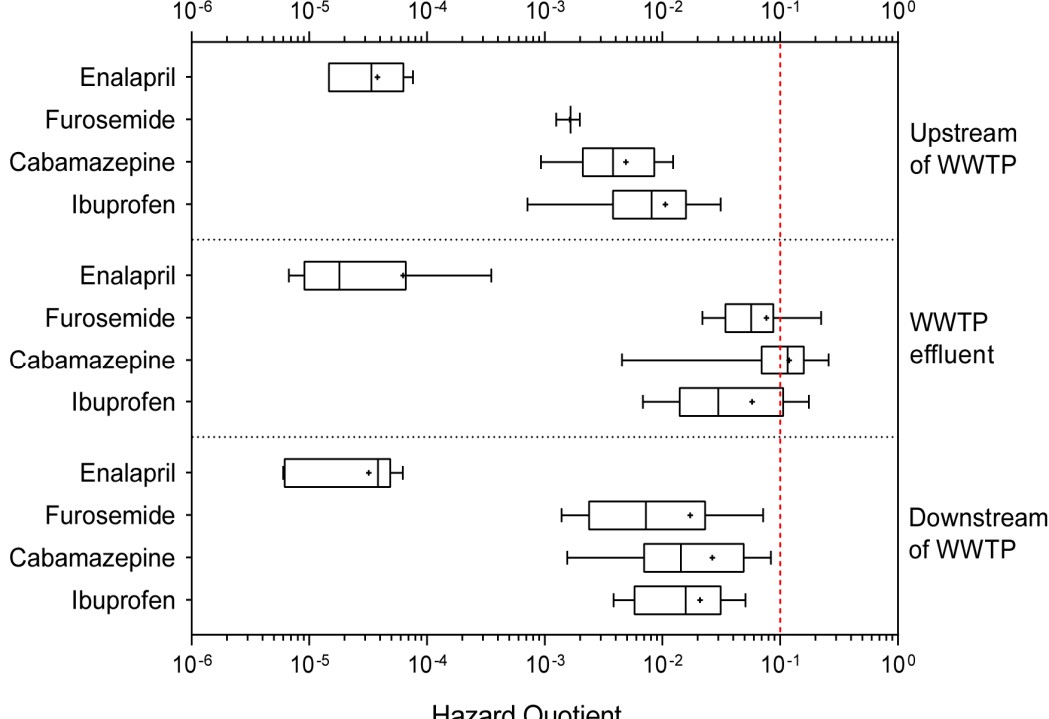

**Figure 4.** Chart representing the Hazard Quotient (HQ) of the investigated pharmaceuticals. The results are plotted on a base 10 logarithmic scale, as box plots with whiskers at the minimum and maximum values, while the box is delimited by the 25th and 75th percentile, the median is shown as a line and the mean as a plus sign. The red dotted line represents the limit of 0.1 HQ.

## 4. Discussion

In the Water Framework Directive (WFD) [66], there are two types of water environmental quality standards (EQS) concerning pharmaceuticals: the annual average concentration (AA-EQS) based on chronic toxicity data and the maximum acceptable concentration (MAC-EQS) [67]. These values are 0.5 μg/L (AA-EQS) and 1600 μg/L (MAC-EQS) for carbamazepine [67] and 1 μg/L (AA-EQS) and 40 μg/L (MAC-EQS) for ibuprofen [68]. There are no values calculated for enalapril, enalaprilat, furosemide, 1-hydroxyibuprofen, 2-hydroxyibuprofen, and carboxyibuprofen. We found the maximum concentration of carbamazepine in the downstream sample to be above the annual average environmental quality standard but lower than the maximum accepted. However, the situation worsens in the WWTP effluent, where the concentration average is higher than the AA-EQS for carbamazepine. Because the average and median concentrations for carbamazepine in the downstream sample are less than half of the average accepted value, these concentrations do not pose a threat as of yet. Still, they should be monitored closely so as to not rise above the annual accepted average through accumulation.

We found that the HQ is lower than one for all the pharmaceuticals tested, in all the sampling sites, both in the WWTP effluent and in the river. The values we found are similar to values already reported [19] or slightly lower than those values. Ibuprofen has been found to have the highest HQ in the WWTP [19], while in our study, furosemide and carbamazepine seem to have slightly higher HQs. The HQ values that range from 0.1 to 1 are considered low hazard with potential adverse effects; between 1 and 10, the adverse effects and hazard are probable, while for values higher than 10, hazard are anticipated [19,69,70]. These results imply that for the pharmaceuticals studied, the measured levels pose a threat to the environment, especially for the effluent. Although WWTPs do not have designated methods for removal of the studies pharmaceuticals, it has been reported that some of them degrade during wastewater treatment. Using reported percentages of WWTP clean-up of pharmaceuticals, 42% for furosemide [55], 20% for carbamazepine [53], 80% for ibuprofen [71], and 95% for enalapril, we calculated the putative influent concentration (Figure 5). The results show much higher concentrations in the influent compared with the effluent (Figure 5). These concentrations could be potentially hazardous, especially during heavy rain periods when WWTPs overflow and discharge the effluent at a higher rate than normal.

Moreover, we considered the impact of these high concentrations of pharmaceuticals on the bacteria contained in the activated sludge and the possibility that they would be killed or inhibited. When comparing the MECs of pharmaceuticals in the WWTP with known concentrations that impact bacterial survival, we find them to be at least three orders of magnitude lower than concentrations that would impact bacterial survival [72]. Even if we extrapolate the putative influent concentrations, we do not obtain values over the risk concentrations. Enalapril, which had the highest degradation/removal rate, has a putative concentration in the influent lower than the rest of the pharmaceuticals investigated. Therefore, we do not anticipate a negative impact caused by this singular pharmaceutical's presence on the biological processes taking place in the WWTP.

Our measurement that in the majority of cases the downstream contamination is lower than the effluent contamination, coupled with the correlations observed between downstream and effluent concentration, leads us to believe that the WWTPs are primary sources of river pollution with pharmaceuticals. The underlying concentration that appears in the river (upstream sites) is not high enough to hint at other sources of contamination that could topple the effect of the WWTPs. The fact that enalaprilat and furosemide concentrations are correlated between the upstream and downstream of WWTP sites could mean that these compounds could quickly traverse the river's length between the WWTP sites before precipitating out of the solution or being degraded. This is backed up by their higher solubility in water compared to the other pharmaceuticals, calculated through the Log $K_{ow}$ parameter (Table 1). The fact that the concentrations of ibuprofen and 2-hydroxyibuprofen are positively correlated in all the investigated cases (upstream, downstream, and WWTP effluent), while concentrations of ibuprofen and carboxyibuprofen are positively correlated

in two cases (upstream and downstream) could be due to the metabolization of ibuprofen in humans, for which 2-hydroxyibuprofen and carboxyibuprofen are final products and 1-hydroxyibuprofen is an intermediary step to carboxyibuprofen [73]. Therefore, it is conceivable that 2-hydroxyibuprofen and carboxyibuprofen would have higher concentrations than 1-hydroxyibuprofen, which appears to be the case (Table 4).

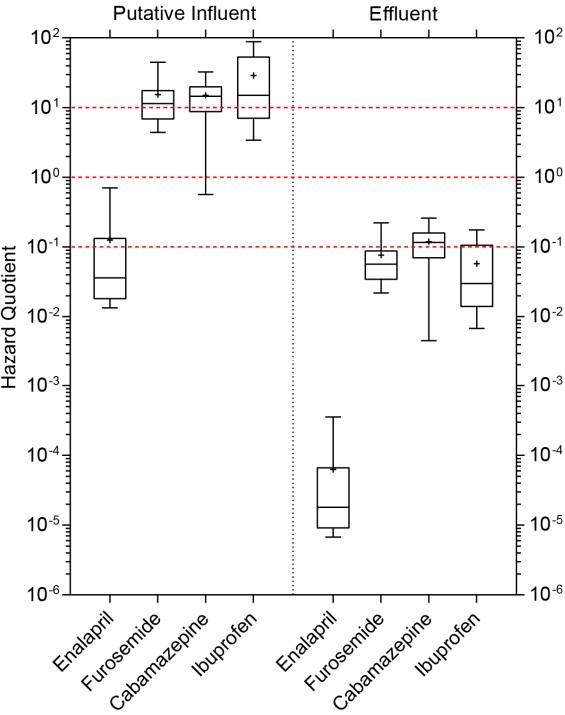

**Figure 5.** Estimation of the influent concentration of pharmaceuticals in the WWTP based on the degradation/removal rate. The results are plotted on a base 10 logarithmic scale as box plots with whiskers at the minimum and maximum values, while the box is delimited by the 25th and 75th percentile, with the median shown as a line and the mean as a plus sign. The red dotted lines represent the limits of 0.1, 1, and 10 HQ.

The frequency of detection was above 50% both in the WWTP effluent and downstream from it, reaching 100% for some of the pharmaceuticals (Table 4). Enalapril and its metabolite enalaprilat had the lowest frequency of detection overall. The rest had low frequencies in the upstream samples, with the frequencies rising above 80% for most pharmaceuticals in the WWTP effluent and downstream from it. Carbamazepine had the highest detection level, with 93% upstream, 93% downstream, and 100% at the WWTP. This high frequency of detection points to several things: it shows the level of pollution with the tested pharmaceuticals, it confirms our ability to measure the presence of these pharmaceuticals in the river water, and it shows the slow degradation rate and persistence of this pharmaceutical in the environment [74].

**Table 4.** Summary of pharmaceutical concentrations measured upstream, downstream, and at the WWTP effluent.

| Location | Substance | Min (ng/L) | Max (ng/L) | Median (ng/L) | Average (ng/L) | N | F |
|---|---|---|---|---|---|---|---|
| Upstream | Enalaprilat | 1.19 | 23.16 | 7.66 | 9.02 | 6 | 40% |
| | Enalapril | 2.70 | 14.00 | 6.22 | 6.96 | 5 | 33% |
| | Furosemide | 7.78 | 12.30 | 10.27 | 10.12 | 3 | 20% |
| | Carbamazepine | 7.16 | 95.11 | 29.38 | 37.72 | 14 | 93% |
| | Carboxyibuprofen | 6.74 | 111.61 | 26.30 | 31.30 | 13 | 87% |
| | 2-hydroxyibuprofen | 2.27 | 49.49 | 2.27 | 2.27 | 15 | 100% |
| | 1-hydroxyibuprofen | 0.28 | 4.66 | 2.19 | 2.39 | 13 | 87% |
| | Ibuprofen | 1.65 | 71.85 | 18.59 | 24.39 | 12 | 80% |
| WWTP effluent | Enalaprilat | 2.54 | 146.66 | 7.93 | 24.82 | 9 | 60% |
| | Enalapril | 1.24 | 64.93 | 3.31 | 11.54 | 9 | 60% |
| | Furosemide | 135.92 | 1379.37 | 352.43 | 473.62 | 14 | 93% |
| | Carbamazepine | 34.95 | 1992.43 | 893.94 | 918.99 | 14 | 93% |
| | Carboxyibuprofen | 17.84 | 2036.70 | 91.36 | 431.01 | 12 | 80% |
| | 2-hydroxyibuprofen | 15.94 | 826.16 | 85.29 | 221.64 | 12 | 80% |
| | 1-hydroxyibuprofen | 1.30 | 84.22 | 7.82 | 22.85 | 13 | 87% |
| | Ibuprofen | 15.68 | 403.06 | 68.66 | 132.62 | 8 | 53% |
| Downstream | Enalaprilat | 1.83 | 19.07 | 8.83 | 8.90 | 8 | 53% |
| | Enalapril | 1.11 | 11.50 | 7.07 | 5.90 | 7 | 47% |
| | Furosemide | 8.63 | 444.63 | 44.84 | 106.81 | 12 | 80% |
| | Carbamazepine | 11.99 | 643.31 | 110.26 | 204.15 | 15 | 100% |
| | Carboxyibuprofen | 6.76 | 166.28 | 22.38 | 51.74 | 15 | 100% |
| | 2-hydroxyibuprofen | 11.92 | 85.68 | 33.23 | 43.28 | 15 | 100% |
| | 1-hydroxyibuprofen | 0.16 | 11.24 | 2.80 | 4.14 | 14 | 93% |
| | Ibuprofen | 8.84 | 117.14 | 36.17 | 47.88 | 11 | 73% |

N—number of occurrences. F—frequency of detection.

**Table 5.** Concentrations of pharmaceuticals in different locations along the Mureș River Basin.

| | Location | ENA | ENP | FUR | CBZ | CarboxyIBP | 2-hydroxyiIBP | 1-hydroxyIBP | IBP |
|---|---|---|---|---|---|---|---|---|---|
| 1 | Upstream | 4.88 | NA | NA | 20.32 | 6.74 | 21.57 | NA | NA |
| | Downstream | 7.13 | NA | NA | 22.45 | 6.79 | 27.56 | NA | NA |
| | Effluent | 10.55 | 3.31 | 150.26 | 444.73 | 25.19 | 548.05 | 6.99 | 271.60 |
| 2 | Upstream | NA | NA | NA | 17.47 | NA | 2.89 | 2.86 | NA |
| | Downstream | NA | NA | 68.65 | 477.26 | 14.90 | 29.04 | 2.84 | NA |
| | Effluent | 4.32 | NA | 135.92 | 1224.15 | 25.20 | 69.87 | 3.07 | NA |
| 3 | Upstream | NA | NA | NA | 30.25 | NA | 32.10 | 0.28 | 1.65 |
| | Downstream | NA | NA | 143.36 | 377.68 | 93.31 | 74.65 | 0.16 | NA |
| | Effluent | NA | NA | 441.74 | 1230.86 | 254.44 | 226.54 | 1.30 | NA |
| 4 | Upstream | NA | NA | 12.30 | 95.11 | 13.24 | 22.09 | 2.47 | 9.93 |
| | Downstream | NA | NA | 21.04 | 146.56 | 14.18 | 27.44 | 2.79 | 14.07 |
| | Effluent | NA | NA | 298.53 | 1079.31 | 391.16 | 320.95 | 27.08 | 43.14 |
| 5 | Upstream | 1.73 | 2.71 | NA | 8.97 | 27.96 | 43.24 | 1.77 | 24.00 |
| | Downstream | 1.83 | 2.61 | 294.59 | 254.64 | 6.76 | 56.40 | 11.24 | 63.69 |
| | Effluent | NA | 3.20 | 847.10 | 568.59 | 37.55 | 84.94 | 21.94 | 92.47 |
| 6 | Upstream | NA | NA | 7.78 | 71.37 | 15.34 | 21.97 | 2.78 | 15.63 |
| | Downstream | NA | NA | 8.63 | 85.65 | 13.57 | 17.85 | 2.44 | 16.35 |
| | Effluent | 6.51 | NA | 214.79 | 1701.62 | 22.72 | 15.94 | 3.98 | 15.68 |
| 7 | Upstream | NA | NA | NA | 71.23 | 7.57 | 15.77 | 2.16 | 8.32 |
| | Downstream | NA | NA | 14.44 | 110.26 | 8.39 | 16.81 | 2.54 | 8.84 |
| | Effluent | NA | NA | 241.95 | 876.52 | 17.84 | 32.34 | 8.53 | 28.67 |
| 8 | Upstream | 23.16 | 14.00 | NA | 12.59 | 53.54 | 30.13 | 2.55 | 44.05 |
| | Downstream | 19.07 | 11.50 | NA | 94.46 | 166.28 | 85.68 | 9.02 | 107.35 |
| | Effluent | NA | NA | NA | NA | 1646.56 | NA | 84.22 | 403.06 |
| 9 | Upstream | 12.73 | 9.17 | 10.27 | 28.52 | 48.25 | 49.49 | 4.23 | 71.85 |
| | Downstream | 13.34 | 8.93 | 15.91 | 88.20 | 50.50 | 65.68 | 4.55 | 64.87 |
| | Effluent | 7.93 | 7.29 | 325.29 | 1194.42 | NA | 85.64 | 2.22 | 44.84 |
| 10 | Upstream | 10.45 | 6.22 | NA | 7.16 | 26.30 | 37.13 | 2.19 | 36.95 |
| | Downstream | 13.25 | 7.07 | 17.11 | 53.88 | 22.38 | 42.91 | 2.68 | 36.17 |
| | Effluent | 31.17 | 17.04 | 1379.37 | 1992.43 | 542.88 | 321.89 | 61.52 | NA |

**Table 5.** *Cont.*

|    | Location   | ENA   | ENP   | FUR     | CBZ    | CarboxyIBP | 2-hydroxiIBP | 1-hydroxyIBP | IBP    |
|----|------------|-------|-------|---------|--------|------------|--------------|--------------|--------|
| 11 | Upstream   | NA    | NA    | NA      | 63.70  | 33.74      | 31.19        | 1.62         | 15.26  |
|    | Downstream | 3.15  | 1.14  | 444.63  | 643.31 | 25.60      | 33.23        | 7.00         | NA     |
|    | Effluent   | 3.65  | 1.24  | 439.93  | 663.75 | NA         | NA           | NA           | NA     |
| 12 | Upstream   | NA    | 2.70  | NA      | NA     | 8.10       | 2.27         | NA           | NA     |
|    | Downstream | 10.52 | 8.97  | 11.21   | 11.99  | 161.05     | 68.29        | 1.39         | 117.14 |
|    | Effluent   | 146.6 | 64.93 | 207.04  | 225.99 | NA         | NA           | NA           | NA     |
| 13 | Upstream   | 1.19  | NA    | NA      | 38.64  | 45.85      | 26.10        | 1.66         | 21.54  |
|    | Downstream | 2.93  | 1.11  | 101.57  | 234.56 | 29.76      | 11.92        | 2.81         | 13.33  |
|    | Effluent   | 10.04 | 3.49  | 414.28  | 717.23 | 26.65      | 61.82        | 7.82         | NA     |
| 14 | Upstream   | NA    | NA    | NA      | 25.61  | 111.61     | 40.23        | 4.66         | 35.12  |
|    | Downstream | NA    | NA    | NA      | 32.43  | 20.47      | 28.60        | 2.26         | 13.09  |
|    | Effluent   | NA    | 1.27  | 1154.93 | 34.95  | 145.17     | 65.59        | 3.97         | NA     |
| 15 | Upstream   | NA    | NA    | NA      | 37.14  | 8.64       | 10.74        | 1.80         | 8.34   |
|    | Downstream | NA    | NA    | 140.53  | 428.90 | 142.12     | 63.09        | 6.25         | 71.84  |
|    | Effluent   | 2.54  | 2.08  | 379.57  | 911.36 | 2036.70    | 826.16       | 64.42        | 161.47 |

1—Toplița, 2—Reghin, 3—Cristești, 4—Luduș, 5—Câmpia Turzii, 6—Ocna Mureș, 7—Aiud, 8—Sighișoara, 9—Mediaș, 10—Târnăveni, 11—Alba Iulia, 12—Orăștie, 13—Deva, 14—Lipova, 15—Arad. The concentrations are reported in ng/L. NA—under the limit of quantification. ENA—enalaprilat. ENP—enalapril, FUR—furosemide. CBZ—carbamazepine. IBP—ibuprofen.

There is no apparent rise in concentrations from the source of the river towards the exit from Romania to Hungary, which could be the result of a high rate of precipitation out of the solution due to the low level of solubility of the investigated pharmaceuticals. Another reason could be the short half-life of these pharmaceuticals, which are hours to days for ibuprofen [75] and 63 days [76] or 38 days [77] for carbamazepine; this could be an indication of the high degradation rate of these compounds. One way of identifying the accumulation of pharmaceuticals in the river is to look at the river sediment, which could be an interesting topic for future research.

In Romania, the concentration of ibuprofen has been reported between 61.3 and 115.2 ng/L in the Someș River in 2006 [57], between 9 and 63 ng/L at different main localities in the Someș River in 2007 [58], and between 16 and 63 ng/L at the WWTP effluent in the Someș River in 2008 [59]. Concentrations were in the range of 1.65–71.85 ng/L for the upstream sites, 15.68–403.06 ng/L for WWTPs, and 8.84–117.14 ng/L for the downstream sites (Table 4). For carbamazepine, it has been reported that the concentrations ranged from 67 to 75 ng/L in the Someș River in 2006 [57] and 38 to 56 ng/L in 2008 [58] and 20–49 ng/L in the Danube River, while a maximum concentration of 140 ng/L was detected in the Argeș River [76]. In 2015, it was also reported that the concentration of carbamazepine was situated in the interval of 4 to 40 ng/L in the Danube River and some tributaries [61] and the interval of 5 to 25 ng/L for some major Romanian rivers (Prahova, Timiș, Danube, Siret, Prut, and Jijia) [60]. The previously reported concentrations have a maximum that is lower than the concentrations reported in this study, at all the investigated sites (upstream, WWTP, and downstream) (Table 4). The only exception is for the concentration of ibuprofen from the Târgu Mureș WWTP, which was investigated in three studies [36–38] and have identified high concentrations, up to 7600 ng/L. These results point to a problematic situation of the Mureș River as a hotspot, with levels of carbamazepine in the WWTP effluent reaching a maximum of 1992.43 ng/L. These high concentrations could be due to sampling the effluent, where higher concentrations are to be expected. Another explanation for these high MECs is the fact that Mureș River has a smaller volume of water than the Danube, which makes detecting higher concentrations of pharmaceuticals more likely, as already pointed out [60]. The fact that we detected these pharmaceuticals in high concentrations does not come as a surprise, especially given that a European wide study in 2009 has detected ibuprofen and carbamazepine, among others, as compounds with the highest maximum concentrations in the range of μg/L [77]. Taking into consideration the 90th percentile and the proposed indicative warning levels mentioned in 2009 [77], we can see that the concentrations that we detect are well above the threshold for carbamazepine (limit of 100 ng/L) when looking at the average concentrations of all the substances detailed in this paper in the WWTP effluent and downstream of the WWTP, and ibuprofen (limit of 200 ng/L) in the WWTP

effluent when looking at the maximum detected concentration. In this respect, the Mureș River Basin can be considered polluted, and these results warrant further investigation.

The potential synergic effects of these pharmaceutical compounds with other pollutants present in the basin, such as POPs [12,13,78], can raise the area's risk potential for natural and semi-natural ecosystems and human settlements health and welfare, which increases the importance of monitoring in the catchment basin of rivers receiving wastewater [79,80]. The problem with high levels of pharmaceuticals in the environment is tied directly to human exposure. In this case, the concentrations to which we are exposed could be higher, especially when the individuals are under treatment with the investigated pharmaceuticals.

The potential risk associated with these pharmaceuticals can rise in the present situation. The climate changes [81] tend to reduce the minimum, average, and maximum dilution flow [5] in the Danube Basin too, and the increasing human water consumption will put supplementary pressure on this respect as well.

## 5. Conclusions

Based on this study's results, the large and complex hydrographical system Mureș River Basin was transformed from a grey area to a hotspot in contamination with emerging pharmaceuticals. Pharmaceutical concentrations were found to be the highest in WWTP effluents, indicating that the treatment plants are the primary source of water pollution in this case. The detected levels are higher than the safety limit for carbamazepine in the annual average environmental quality standards, and are higher than the limits proposed in 2009 [77] for carbamazepine and ibuprofen. The determined HQ values imply that the measured levels for the studied pharmaceuticals do pose a threat to the environment.

Pharmaceuticals have become a norm in every household, and unfortunately, this has resulted in the highlighted contamination problem for environmental waters. Since pharmaceuticals are necessary for human and household animal health, and their consumption cannot be eliminated, there should be methods put in place that take care of their disposal, release into the environment, and ecotoxicological effects. Among these methods, an organized monitoring system for out of date pharmaceuticals, a designated treatment at the WWTPs for degradation of these pharmaceuticals, introduction of phytoremediation, depuration stations along the river, and better human population education for the use and disposal of pharmaceuticals should be a priority.

**Author Contributions:** All the authors (A.B., I.B., C.-M.M., D.B., C.M. and A.C.-B.) contributed to the data analyses and writing of the paper; all the authors have read and agreed to the paper's content. D.B. and A.C.-B. contributed to the fieldwork; I.B. and C.-M.M. did the laboratory work; A.B. and A.C.-B. equally contributed to the paper with paper ideas, data analyses, and writing. The corresponding author agreed to cover, full or in part, the article processing charge. All authors have read and agreed to the published version of the manuscript.

**Funding:** This work was supported by the National Council for Higher Education Financing ["Cercetare de excelență în ecotoxicologia poluanților emergenți", CNFIS-FDI-2018-0317].

**Acknowledgments:** We express our gratitude for funding to the National Council for Higher Education Financing "Cercetare de excelență în ecotoxicologia poluanților emergenți," CNFIS-FDI-2018-0317, "Lucian Blaga" University of Sibiu, Applied Ecology Research Center, and Hasso Plattner Foundation research action LBUS-RC-2020-01-12. We would also like to express our gratitude to Florin Albu, for his help in developing the method for detection and quantification of pharmaceuticals from river water.

**Conflicts of Interest:** The authors declare no conflict of interest.

**Declarations:** *Ethics approval and consent to participate:* The manuscript is original, has not been published, and is not currently under consideration by another journal. All the authors of the submitted manuscript respect all the professional and editorial ethic general rules and agree to the terms of the Springer Open License Agreement. *Consent for publication:* All the authors of the submitted manuscript consent to participate as co-authors. *Availability of data and materials:* Any data related to the paper manuscript are available on request; the correspondence author can send the requested data.

## Abbreviations

| | |
|---|---|
| HQ | Hazard Quotient |
| WWTP | Wastewater Treatment Plants |
| PNEC | Predicted No Effect Concentration |
| MEC | measured environmental concentration |
| CAS | Chemical Abstracts Service |
| HPLC | High-Performance Liquid Chromatography |
| SPE | Solid Phase Extraction |
| RT | Retention Times |
| QqQ MS | Triple Quadrupole Mass Spectrometry |
| ESI | Electrospray Ionization |
| APCI | Atmospheric Pressure Chemical Ionization |
| CID | Collision Induced Dissociation |
| MRM | Multiple Reaction Monitoring |
| Q | Quantification Ion |
| q | Qualification Ion |
| LOD | Limit of Detection |
| LOQ | Limit of Quantification |
| RSD | Relative Standard Deviation |
| WFD | Water Framework Directive |
| EQS | Environmental Quality Standards |
| AA-EQS | Annual Average Concentration |
| MAC-EQS | Maximum Acceptable Concentration |
| ENA | Enalaprilat |
| ENP | Enalapril |
| FUR | Furosemide |
| CBZ | Carbamazepine |
| IBP | Ibuprofen |
| POPs | Persistent Organic Pollutants |

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
