# Peer review of "Adding the Mureş River Basin (Transylvania, Romania) to the List of Hotspots with High Contamination with Pharmaceuticals"

_sustainability, doi:10.3390/su122310197_

Round 1

Reviewer 1 Report

Overview and general recommendation:
Biomonitoring studies are constantly being developed field of research. More and more diagnostic tools are being proposed and monitoring of various pharmaceuticals is part of this trend. The authors have made an attempt to a comprehensive comparison of three points potentially affected by pharmaceuticals contamination at 15 sampling sites in the Mureş River Basin. The aim of the study is well-founded and the results constitute a good start for further research considering i.e. aquatic organisms. The manuscript requires minor revision and I recommend its publication in the Sustainability journal.

Abstract
Lines 25-27: This sentence is quite confusing, I suggest to split it into 2 shorter ones.
Lines 29-32: This sentence presents a conclusion, thus I suggest to shift it to this section of the Abstract.
Lines 34-36: This sentence suits better the Background section.
Line 39: I suggest to change ‘measured‘ to ‘recorded‘.
Line 40: I suggest to change ‘highest average measured‘ to ‘highest mean values‘.
Line 43: ‘were‘ instead ‘are‘.
Line 53: The primary sources would rather be the one mentioned in lines 153-155. Thus, WWTPs would be secondary sources.
Line 54: ‘were‘ instead ‘are‘.

Keywords: rivers are not necessary, this word was used in the title (in singular form).

Background
Lines 62-75: Although this fragment is quite interesting, it is very general and is not relevant to the study. I suggest to remove it and start from line 76. It is also a source of 16 references.
Lines 82-85: Such a sentence would be also valuable in the background section in the abstract.
Line 86: Add ‘In‘ at the beginning.
Lines 92-96: This is a detailed description of the sampling site, which should be included in the material and methods section.
Line 105: delete present.
Line 129: mammals include humans, thus I suggest to rewrite this part of the sentence. A group of fish could be added as examples are presented in the next 2 sentences.

Material and methods
Lines 184: Add at least the year of sampling.
Lines 185-188: All this information are included in figure 1, I suggest to delete this repetition from the text.
Sections 2.3.-2.5: I suggest to add some references to the method used.
Line 281: ‘spike‘ instead of ‘spikes‘.
Line 289: delete a dot after ‘extraction‘.

Results

Tables: All tables are cited in the text, however, probably during pdf proof creation, table 4 was cut at the downstream results.

Figures: All figures were cited in the text.
Fig. 2. Symbols description is provided in the figure title, thus the additional legend is unnecessary. I also suggest to add (in the figure title) ‘effluent‘ after WWTP.
Fig. 5. In the title, change ‘As‘ into ‘as‘ and add coma after ‘values‘.

Discussion
Lines 360-361: This information was provided in lines 101-102 and does not introduce into the discussion, thus I suggest to remove it.
Lines 362: Add the citation of WFD (and also supply references).
Lines 384-386: These are the results of the study, which should be presented in the results section. Also figure 6 should be included in the results section, not discussion.
Line 396: Remove figure 6 citations.
Line 400: As I mentioned above (abstract, comment to line 53), the Authors should reconsider if the WWTP is the primary source of pollution as they mention various sources of pharmaceuticals in the aquatic environment before they reach WWTP (in lines 153-155).
Line 404: ‘compared to‘ instead of ‘instead of‘.
Line 493: ‘detected‘ instead of ‘detect‘.

Conclusions
Line 497: See the remark to lines 53 and 400.

References: Positions from 33 to 38 are not cited in the text.

Author Response

Dear Editors and reviewer 1,

We received the comments and suggestions of the reviewers that evaluated our manuscript entitled: Adding the Mureş River Basin (Transylvania, Romania) to the list of hotspots with high contamination with pharmaceuticals, authors: Alexandru Burcea, Ioana Boeraş, Claudia-Maria Mihuţ, Doru Bănăduc, Claudiu Matei and Angela Curtean-Bănăduc. Thank you for editors and reviewers supportive work and help.

Below you will find our response to Reviewer 1.

Reviewer 1

Comment: Lines 25-27: This sentence is quite confusing, I suggest to split it into 2 shorter ones.

Response: The sentence was split into two sentences for more clarity. Lines: 23-25.

Comment: Lines 29-32: This sentence presents a conclusion, thus I suggest to shift it to this section of the Abstract.

Response: The sentence was moved to the conclusion section of the abstract. Lines: 49-51.

Comment: Lines 34-36: This sentence suits better the Background section.

Response: We have clarified the sentence so that it clearly states that we were referring to our study results. Lines: 30-32.

Comment: Line 39: I suggest to change ‘measured‘ to ‘recorded‘.

Response: We have made the change. Line: 34.

Comment: Line 40: I suggest to change ‘highest average measured‘ to ‘highest mean values‘.

Response: We have made the change. Lines: 35-36.

Comment: Line 43: ‘were‘ instead ‘are‘.

Response: We have made the change. Line: 38.

Comment: Line 53: The primary sources would rather be the one mentioned in lines 153-155. Thus, WWTPs would be secondary sources.

Response: We agree that the mentioned sources are important but from our study (the fact that the highest concentration is always at the WWTP and the second highest is the downstream site) it is clear that the WWTP is the source. The fact that concentrations are to be found in the upstream site is most probably due to some upstream WWTP site or indeed the other sources but because the corelation is between WWTP contaminant and the downstream site we concluded that this is the major source of contamination. If other sources would be present, that had higher impact it would destabilize the corelation. We therefore made no change in the manuscript regarding this comment.

Comment: Line 54: ‘were‘ instead ‘are‘.

Response: We have made the required change. Lines: 47-48.

Comment: Keywords: rivers are not necessary, this word was used in the title (in singular form).

Response: We deleted the word “rivers” from keywords.

Comment: Lines 62-75: Although this fragment is quite interesting, it is very general and is not relevant to the study. I suggest to remove it and start from line 76. It is also a source of 16 references.

Response: We agree that the fragment did not add valuable information to the article and so we have removed it.

Comment: Lines 82-85: Such a sentence would be also valuable in the background section in the abstract.

Response: We agree that this is an important addition to the background section of the abstract. We have added a sentence to that accord. Lines: 25-27.

Comment: Line 86: Add ‘In‘ at the beginning.

Response: We agree that this was necessary. We have made the modification. Line: 64.

Comment: Lines 92-96: This is a detailed description of the sampling site, which should be included in the material and methods section.

Response: We agree that the description of the sampling site should be more detailed. We have added information about the sampling sites and kept the fragment in the background section as an introductory phrase about the chosen region of study. We have also added GPS coordinated to figure 1 . Lines: 154-157 and 164-172.

Comment: Line 105: delete present.

Response: We have clarified the sentence by removing the word present.

Comment: Line 129: mammals include humans, thus I suggest to rewrite this part of the sentence. A group of fish could be added as examples are presented in the next 2 sentences.

Response: We agree with this observation and made the necessary modifications. Lines: 104-107.

Comment: Lines 184: Add at least the year of sampling.

Response: We have added the sampling year. Lines: 154-155.

Comment: Lines 185-188: All this information are included in figure 1, I suggest to delete this repetition from the text.

Response: We agree that the information is doubled therefore we have deleted the repetition from the text.

Comment: Sections 2.3.-2.5: I suggest to add some references to the method used.

Response: We have added references to the methods that were used. Lines:185, 196, 209.

Comment: Line 281: ‘spike‘ instead of ‘spikes‘.

Response: We have modified the word to spiked so that it makes sense in the sentence. Lines: 251.

Comment: Line 289: delete a dot after ‘extraction‘.

Response: We have corrected the mistake. Lines: 258.

Comment: All tables are cited in the text, however, probably during pdf proof creation, table 4 was cut at the downstream results.

Response: We have formatted Table 4 and Table 5 in order to fit the width of one page. We also modified the other tables to keep with the theme of table 4 and 5.

Comment: Fig. 2. Symbols description is provided in the figure title, thus the additional legend is unnecessary. I also suggest to add (in the figure title) ‘effluent‘ after WWTP.

Response: We have removed figure 2 because reviewer 3 deemed it not necessary since the same information can be found in table 4 and 5.

Comment: Fig. 5. In the title, change ‘As‘ into ‘as‘ and add coma after ‘values‘.

Response: We agree with the proposed changes. We have made the modifications. Lines: 316-319.

Comment: Lines 360-361: This information was provided in lines 101-102 and does not introduce into the discussion, thus I suggest to remove it.

Response: We agree with the reviewers assessment and have removed the sentence.

Comment: Lines 362: Add the citation of WFD (and also supply references).

Response: We have added the citation for WFD and supplied references for the different values presented for MAC-EQS and AA-EQS. Lines: 322.

Comment: Lines 384-386: These are the results of the study, which should be presented in the results section. Also figure 6 should be included in the results section, not discussion.

Response: We disagree with the reviewer since the clean-up percentages are from literature and were interpreted by us to compute figure 6. Therefore, this is an adaptation of results found in literature with our own interpretation and our own data intermingled. We have not made the change and we hope that this explanation covers why we did not.

Comment: Line 396: Remove figure 6 citations.

Response: We have removed the figure 6 citations.

Comment: Line 400: As I mentioned above (abstract, comment to line 53), the Authors should reconsider if the WWTP is the primary source of pollution as they mention various sources of pharmaceuticals in the aquatic environment before they reach WWTP (in lines 153-155).

Response: We consider that based only on our results there is no evidence that we can use to say that there are other primary sources. What the reviewers suggest is not possible to conclude from what we have measured. For the samples we have collected and based on the fact that the downstream concentration is always corelated with the effluent concentration and always smaller we concluded that the WWTPs are the sources of contamination. The presence of pharmaceuticals in upstream water does not change the correlation between WWTP and downstream which would have happened if there were other sources with higher capacity of contamination than WWTPs. We have added this discussion to the article so that we clearly state our findings and interpretation of said findings. Lines: 354-358.

Comment: Line 404: ‘compared to‘ instead of ‘instead of‘.

Response: We have made the necessary modifications. Lines: 360-362.

Comment: Line 493: ‘detected‘ instead of ‘detect‘.

Response: We have made the necessary modifications. Lines: 442-443.

Comment: Line 497: See the remark to lines 53 and 400.

Response: We have discussed this comment in a previous response. We have added the necessary information to the article to make our position clear. Lines: 354-358.

Comment: Positions from 33 to 38 are not cited in the text.

Response: These references are cited in Table 1. Lines: 94-99.

Reviewer 2 Report

This paper is important addition to our knowledge of riverine impacts, the emerging pharmaceutical pollutants is most pertinent to these times.  I am not a chemist so cannot comment on the laboratory methods, but as a freshwater riverine ecologist I feel I can review the main thrust of the paper. Technically the paper is very good, but the English needs more work. If I had a word document that I could use track changes I could make many suggestions but feel it is not my role as reviewer to do so. There is a lot of variability with present and past tense used interchangeably this needs to be consistent.  A good review and rewrite by an English speaker would much improve the paper.

I was left with questions that were unanswered in the paper; Why are Romanian WWTP pharmaceutical levels higher than the rest of Europe? Is the list of pharmaceuticals tested for in this paper a small or comprehensive subset of the emerging pharmaceuticals? What I mean is; is this the tip of the iceberg or all the likely contenders?

Specific comments:

The abstract contains abbreviations that are seen for the first time for reader so should be in full.

Line 50-51. The “European map” is referred to, but not specified, is this a map of pharmaceutical concentrations in rivers? If so, a reference is needed.

Line 64. The word antique is usually reserved for objects not for civilisations.

Line 109 – The word drugs is used instead of Pharmaceuticals, consistence or clarify the difference between the two.

Table 1 Header abbreviations should be spelt out in caption.

Line 133 “polar nature” is this supposed to be refereeing to molecular weight?

Line 145 “grey spot” does this refer to the mystery European map? Or a grey area as in an “ill-defined situation”

146 you say few reports but then refer to many of them in the reference list

Line 322 Figure 3 caption the colour gradient must be defined; I can guess that dark red colour is high correlation, but this must be clarified

Line 495 the undefined “European map” again

Line 496 “Pharmaceutical concentrations were highest in WWTP effluents” this seems very obvious, but then the next sentences says they were the highest in Europe, “indicating the WWTPs are the primary source” – I can’t follow the logic here, the question for me is why are these Romanian WWTPs so much higher than the rest of Europe and the fact that they are the source in the rivers seems obvious.

Line 553 ‘date’ should be ‘data’

Author Response

Dear Editors and reviewer 2,

We received the comments and suggestions of the reviewers that evaluated our manuscript entitled: Adding the Mureş River Basin (Transylvania, Romania) to the list of hotspots with high contamination with pharmaceuticals, authors: Alexandru Burcea, Ioana Boeraş, Claudia-Maria Mihuţ, Doru Bănăduc, Claudiu Matei and Angela Curtean-Bănăduc. Thank you for editors and reviewers supportive work and help.

Below you will find our response to Reviewer 2.

Comment: There is a lot of variability with present and past tense used interchangeably this needs to be consistent. 

Response: We have addressed some of the variability from the manuscript and tried to keep it consistent. Two native English speakers have reviewed the text.

Comment: I was left with questions that were unanswered in the paper; Why are Romanian WWTP pharmaceutical levels higher than the rest of Europe? Is the list of pharmaceuticals tested for in this paper a small or comprehensive subset of the emerging pharmaceuticals? What I mean is; is this the tip of the iceberg or all the likely contenders?

Response: The reason for which the Romanian WWTP concentrations are higher than other countries from Europe could be that even though there are regulations in place these are not enforced. But since we have no proof of that we did not comment on that matter. The emerging pollutants are ibuprofen and carbamazepine the rest are widely used pharmaceuticals that were tested to see if their concentration exceeds that of ibuprofen and carbamazepine. These are not all the likely contenders – especially since the field of emerging pollutants is broadening each day. We have commented about this in the article so that we clearly state what the situation is. Lines: 144-150.

Comment: The abstract contains abbreviations that are seen for the first time for reader so should be in full.

Response: We have added the abbreviations to the abstract. Lines: 23-28.

Comment: Line 50-51. The “European map” is referred to, but not specified, is this a map of pharmaceutical concentrations in rivers? If so, a reference is needed.

Response: We were referencing an abstract ‘European map’ in terms of articles that have investigated pollutants in different countries from Europe. To avoid any confusion we have removed the term ‘European map’ from the manuscript (title and conclusions). Lines: 1-2.

Comment: Line 64. The word antique is usually reserved for objects not for civilisations.

Response: We have removed the section entirely.

Comment: Line 109 – The word drugs is used instead of Pharmaceuticals, consistence or clarify the difference between the two.

Response: We have replaced the word ‘drug’ with ‘pharmaceutical’ throughout the manuscript.

Comment: Table 1 Header abbreviations should be spelt out in caption.

Response: We have added the abbreviations in the caption. Lines: 94-99.

Comment: Line 133 “polar nature” is this supposed to be refereeing to molecular weight?

Response: No. Compounds can be polar or non-polar (or anything in between because in reality it is a gradient not a black and white situation). Polar compounds dissolve in water. Non-polar compounds do not dissolve in water and adhere to sediment particles. The polar nature of a compound refers to how easily it dissolves in water. And because the studied compounds have LogKow values that are consistent with polar compounds they are slightly soluble in water.

Comment: Line 145 “grey spot” does this refer to the mystery European map? Or a grey area as in an “ill-defined situation”

Response: We were referring to ‘grey area’. We have modified the manuscript so that all the instances where ‘grey spot’ was mentioned now read ‘grey area’. Line: 121.

Comment: 146 you say few reports but then refer to many of them in the reference list

Response: We have found three instances where only one WWTP was investigated (the same in all three articles). We can still say that these are a few articles and the overall situation for the Mureș River Basin was not clear until our study came to be. We have commented on the concentrations that the other authors have found in the region.

Comment: Line 322 Figure 3 caption the colour gradient must be defined; I can guess that dark red colour is high correlation, but this must be clarified

Response: We have added a colour gradient under each graph where 1 is dark red and -1 is dark blue. We have also added that 1 means positive corelation and -1 negative corelation in the caption of the figure. Lines: 285-289 and 301-305.

Comment: Line 495 the undefined “European map” again

Response: We have removed the term altogether to not create any more confusion.

Comment: Line 496 “Pharmaceutical concentrations were highest in WWTP effluents” this seems very obvious, but then the next sentences says they were the highest in Europe, “indicating the WWTPs are the primary source” – I can’t follow the logic here, the question for me is why are these Romanian WWTPs so much higher than the rest of Europe and the fact that they are the source in the rivers seems obvious.

Response: We have clarified the sentence. Lines: 439-445

Comment: Line 553 ‘date’ should be ‘data’

Response: We have corrected the mistake.

Reviewer 3 Report

Comment 1:

Aim

The aim category of the article is too long and should be shortened. The aim needs to be more concise and clear.

Comment 2:

  1. Material and methods

2.1. Sampling

You have added the names of the places where WWTPs are found but there is no data about the GPS location. I think this is of importance for readers to be able to pinpoint the problematic areas.

Comment 3:

  1. Material and methods

2.1. Sampling

You have mentioned that plastic bottles were used for sampling (“. Water was collected in plastic (polyethylene terephthalate) bottles, capped, and maintained at 4°C for a maximum of two weeks.”). Why is that? Wouldn’t brown glass bottles be better?

Comment 4:

2.5. Triple Quadrupole Mass Spectrometry

Why is there only one ion transition mentioned in Table 2 for Ibuprofen and metabolites? Why is there no qualifier ion mentioned. If this is an omission please add the data.

Comment 5:

2.5. Triple Quadrupole Mass Spectrometry

In Table 3 you have mentioned the MDL (method detection limit) but then in Table 5 you have presented the data as higher than LOQ (limit of quantitation). Since the MDL is lower in some cases than LOQ there is no point to mention it at all. Please remove the information about MDL and its determination.

Comment 6:

2.7. Method validation

It is not clear how the recovery experiment was undertaken. Were there eight blinds and eight equivalent recovery samples. Were the blinds from the same water sample. Please clarify.

Comment 7:

3.1. Concentrations

There is no reason that Figure 2 should exist. The same information could be deemed from Table 4 and Table Table 5 and there you can see the values and future readers could use them in their own research. Please remove Figure 2.

Comment 8:

3.3. Hazard Quotient (HQ)

Even though you have presented the HQ (hazard quotient) values in the figure you have not presented them in text. I think that a table with the HQ values should be given to the readers. This could be done as supplementary. Please add HQ values table.

Comment 9:

  1. Discussion

(“These concentrations remain under the risk concentrations even when we extrapolate the influent concentrations based on the WWTP’s degradation/removal rate (Figure 6). Enalapril, which had the highest degradation/removal rate (Figure 6), has a putative concentration in the influent lower than the rest of the pharmaceuticals investigated.”) These two sentences do not make sense for me. I suggest clarifying them.

Comment 10:

  1. Conclusions

The conclusion section is too long. I suggest shortening the conclusions.

Author Response

Dear Editors and reviewers,

We received the comments and suggestions of the reviewers that evaluated our manuscript entitled: Adding the Mureş River Basin (Transylvania, Romania) to the list of hotspots with high contamination with pharmaceuticals, authors: Alexandru Burcea, Ioana Boeraş, Claudia-Maria Mihuţ, Doru Bănăduc, Claudiu Matei and Angela Curtean-Bănăduc. Thank you for editors and reviewers for their supportive work and help.

Below you will find our response to Reviewer 3.

Comment: The aim category of the article is too long and should be shortened. The aim needs to be more concise and clear.

Response: We have shortened the aim category. Lines: 143-150.

Comment: You have added the names of the places where WWTPs are found but there is no data about the GPS location. I think this is of importance for readers to be able to pinpoint the problematic areas.

Response: We agree and we have added the GPS location for each sampling site in the caption of figure 1. Lines: 163-172.

Comment: You have mentioned that plastic bottles were used for sampling (“. Water was collected in plastic (polyethylene terephthalate) bottles, capped, and maintained at 4°C for a maximum of two weeks.”). Why is that? Wouldn’t brown glass bottles be better?

Response: No. Glass is not recommended for pharmaceutical sampling. Various interactions could arise when pharmaceuticals come in contact with glass surfaces including leaching, ion exchange, precipitation, glass dissolution, etc. We did not add this information in the manuscript so as to not create confusion.

Comment: Why is there only one ion transition mentioned in Table 2 for Ibuprofen and metabolites? Why is there no qualifier ion mentioned. If this is an omission please add the data.

Response: This is not an omission. Ibuprofen and its metabolites are unstable in the Ion Source making them hard to quantify. Most articles that deal with triple quadrupole quantification of ibuprofen are using just one transition without qualifier ion. We therefore chose to do the same.

Comment: In Table 3 you have mentioned the MDL (method detection limit) but then in Table 5 you have presented the data as higher than LOQ (limit of quantitation). Since the MDL is lower in some cases than LOQ there is no point to mention it at all. Please remove the information about MDL and its determination.

Response: We have removed any mention of MDL from the manuscript.

Comment: It is not clear how the recovery experiment was undertaken. Were there eight blinds and eight equivalent recovery samples. Were the blinds from the same water sample. Please clarify.

Response: We have clarified the method validation section. Lines: 247-260.

Comment: There is no reason that Figure 2 should exist. The same information could be deemed from Table 4 and Table Table 5 and there you can see the values and future readers could use them in their own research. Please remove Figure 2.

Response: We have removed figure 2.

Comment: Even though you have presented the HQ (hazard quotient) values in the figure you have not presented them in text. I think that a table with the HQ values should be given to the readers. This could be done as supplementary. Please add HQ values table.

Response: We agree that the values should be made available. We suggest to the editor if it is possible to have the HQ values as a supplementary file. We have provided such a file: Table Supplementary 1. And we have referenced the file in the manuscript. Lines: 313

Comment: (“These concentrations remain under the risk concentrations even when we extrapolate the influent concentrations based on the WWTP’s degradation/removal rate (Figure 6). Enalapril, which had the highest degradation/removal rate (Figure 6), has a putative concentration in the influent lower than the rest of the pharmaceuticals investigated.”) These two sentences do not make sense for me. I suggest clarifying them.

Response: We have clarified the sentences so that it does not create any more confusion. Lines: 349-350.

Comment: The conclusion section is too long. I suggest shortening the conclusions.

Response: We have shortened the conclusions section. Lines: 438-452.